# Large Language Models Miss the Multi-Agent Mark

**Emanuele La Malfa**[1][*]     **Gabriele La Malfa**[2][*]     **Samuele Marro**[3]
**Jie M. Zhang**[2]     **Elizabeth Black**[2]     **Michael Luck**[4]     **Philip Torr**[3]     **Michael Wooldridge**[1]
[1]Department of Computer Science, University of Oxford
[2]Department of Informatics, King's College London
[3]Department of Engineering, University of Oxford
[4]University of Sussex

## Abstract

Recent interest in Multi-Agent Systems of Large Language Models (MAS LLMs) has led to an increase in frameworks leveraging multiple LLMs to tackle complex tasks. However, much of this literature appropriates the terminology of MAS without engaging with its foundational principles. In this position paper, we highlight critical discrepancies between MAS theory and current MAS LLMs implementations, focusing on four key areas: the social aspect of agency, environment design, coordination and communication protocols, and measuring emergent behaviours. Our position is that many MAS LLMs lack multi-agent characteristics such as autonomy, social interaction, and structured environments, and often rely on oversimplified, LLM-centric architectures. The field may slow down and lose traction by revisiting problems the MAS literature has already addressed. Therefore, we systematically analyse this issue and outline associated research opportunities; we advocate for better integrating established MAS concepts and more precise terminology to avoid mischaracterisation and missed opportunities.

## 1   Introduction

The recent machine learning literature has seen an upsurge in popularity of Large Language Models (LLMs) used in coordination to solve complex tasks, a line of research that goes by the name of "Multi-Agent Systems of LLMs" (MAS LLMs) [42, 113]. In MAS LLMs, each LLM-agent specialises in a task to accomplish a goal. A few examples of MAS LLMs use are software engineering [43], multi-robot planning [19], data analysis [92], scientific production, reasoning and debating [29, 111, 141], and social simulations [85], among many others. There is also an increasing interest in open-ended MAS LLMs, systems whose complex interactions give rise to human-like emergent behaviours [3, 38, 85].

However, labelling MAS LLMs as "Multi-Agent Systems" has already raised concerns in the scientific community [17]. Influential frameworks employed in MAS LLMs applications, such as ReAct [134], which in turn leverages methods like Chain of Thought [120], Tree of Thought [133], etc., are single-agent prompting techniques that overlook concurrency and shared states;[2] At the same time, agentic frameworks developed by large companies are monolithic orchestrators that leverage (and only cite) machine learning research, taking little notice of decades of MAS research.[3]

While the term MAS LLMs was introduced in 2023 [110], the first MAS works date back to the 1980s and the 1990s [32, 103, 124]. In this sense, our position advocates for using precise scientific terminology and cautions against the risk of reinventing the wheel. For a relatively new field like that

---

[*]Equal contribution.    Keep the correspondence to `emanuele.lamalfa@cs.ox.ac.uk` and `gabriele.la_malfa@kcl.ac.uk`

[2]`https://gist.github.com/yoavg/9142e5d974ab916462e8ec080407365b`

39th Conference on Neural Information Processing Systems (NeurIPS 2025) Position Paper Track.

of MAS LLMs, failing to engage with the broader MAS literature may lead to overlooked insights and missed opportunities for meaningful advancement.

We articulate our position by identifying three core aspects of MAS that most MAS LLMs in the literature overlook or violate, as well as an aspect related to benchmarking those systems. We criticise the notions of agents' social intelligence and environment as proposed in the MAS LLMs literature (Sections 2 and 3), and discuss what is missing in terms of coordination and communication (Section 4). Further, we observe that the interest in open-ended environments and emergent behaviours is not supported by benchmarks to define, identify or measure such *emergence*; the results are primarily descriptive and risk to over-inflate arguments for LLMs' general and super-intelligence. (Section 5).

We summarise each point below, then state our position.

**I. Social intelligent agents: LLM agents lack native *social behaviour*.** MAS agents populate an environment and receive high-level goals to fulfil. Such goals necessitate the realisation, specification, and completion of other possibly unanticipated sub-tasks. In this context, a high-level goal tests an agent's reactivity, proactiveness, and social abilities [124]. A reactive agent dynamically perceives the environment and takes the initiative to satisfy its design objectives. In doing so, a social agent is capable of interacting and intelligently competing with other agents.

In the context of LLMs, agents are both reactive and proactive: they exhibit remarkable adaptability to changes of an input prompt (which account for their environment, in most cases), and can be trained to take the initiative on how to split a task into its elementary components and then complete them, as shown by frameworks such as AutoGPT [131].

In the current MAS LLM literature, we highlight the lack of consideration given to the social aspects of reactivity and proactiveness. While collaborating and competing are prerequisites of any MAS, most LLMs are fine-tuned as single agents, with no proper multi-agent pre-training procedure [54].[3] In other words, LLMs are trained in isolation to respond to users' requests, rather than to interact with each other. This leads to poor performance and unexpected failures when LLMs are benchmarked to identify other agents' beliefs, desires and intentions, i.e., with Theory of Mind problems [100, 115].

These simplifications limit the field's potential and risk misdirecting current efforts toward MAS, when in some cases the problem may be better addressed by aggregation methods that combine many independent components, such as ensembles [30, 53, 65, 107].[4]

In Section 2, we expand on these arguments.

**II. Environment design: MAS LLMs environments are LLM-centric.** MAS traditionally model the environment with no strong assumption on the architecture or configuration of the agents that populate it [23, 86, 126]. Conversely, MAS LLMs subvert this approach: the environment design assumes the *agents are LLMs* that communicate and coordinate via natural language. This Section warns that this assumption overlooks the intrinsic limitations of LLMs and argues for addressing such problems by designing MAS environments that are not LLM-centric.

Consider the environment predictability: for an environment that is meant to be deterministic, LLMs are inherently not [58, 83]; thus, a fully deterministic environment cannot exist when involving LLMs, providing little control over procedures one wants to guarantee to be safe or terminate in a specific state.

Further, agents should receive and consistently maintain a unique representation of the environment [31, 90], particularly when settings are dynamic and partially observable [81]. To foster effective cooperation and competition, that should not reduce to the environment but include representing other agents' actions, beliefs and intentions. Unfortunately, LLMs are not only known to struggle with inferring beliefs and intentions [97, 115], a long-standing challenge in the MAS literature [93, 102], but also with maintaining a consistent representation of the environment due to issues like hallucinations and memory persistency.

Another critical aspect of MAS is how the environment is represented and then perceived by the agents. In most MAS LLMs settings, the environment is textual or translated as such, with text being

---

[3]https://www.kaggle.com/whitepaper-agents, https://www.anthropic.com/engineering/building-effective-agents, https://openai.com/index/new-tools-for-building-agents/

[4]The authors of "More agents is all you need" make explicit that their technique is an ensemble, as per their Figure 1 [65].

the *medium* of reference for the largest majority of open- and closed-source LLMs. Storing such representations may rapidly exceed a model's context length and cause hallucinations [61, 67].

To realise LLMs as MAS agents that work in coordination with humans and other non-LLM agents, we should design open, multi-modal environments that address their lack of long-term memory persistency [140], non-determinism and propensity to hallucinations [58], and the costs and intrinsic ambiguity of natural language as a storage and communication medium [76].

Section 3 expands on these issues.

**III. Coordination and communication issues in MAS LLMs.** Interaction among agents plays a crucial role in enabling intelligent, decentralised coordination. However, the current MAS LLMs overlook several critical aspects of MAS, including synchronised coordination, concurrent systems, and communication methods.

A prototypical scenario in a concurrent MAS is an agent that processes data generated, at random intervals, by another agent. An error may occur when the first agent receives new data before it has finished processing a previous batch. Standard solutions include storing the data or a mechanism to inform the other system to send the data later.

Asynchronicity is typically absent in MAS LLMs, as LLM-agents often operate in strictly sequential pipelines or parallel, rather than as independent, concurrently operating agents. While some works are moving in this direction [39], we argue for leveraging the body of knowledge from the field of multi-agent concurrent systems that is operative since the 1990s (see, for example, [60, 125]).

Another fundamental aspect of MAS and MAS LLMs is communication between agents. Most MAS LLMs literature assumes agents communicate via natural language [88]. This smoothness is a simplification that overlooks the complexity of real agent interaction and decades of research in MAS communication systems and protocols [11, 48, 96].

Natural language communication is, in fact, inefficient and ambiguous: in contrast, MAS communication protocols in the form of structured languages (e.g., KQML [36] or Agent-Oriented Programming [103]) are consolidated *performative* standards that describe agents' beliefs, commitments, and actions. In line with the MAS literature, we argue mechanisms to negotiate and implement communication methods that integrate the principles of speech acts [6, 104] and Gricean maxims [41] to minimise the cost of communication and maximise its effectiveness.

We expand on these points in Section 4.

**IV. MAS LLMs do not quantify the emergence of complex behaviours.** Several recent works considered MAS LLMs as open-ended systems [3, 85, 128], i.e., scenarios where the long-term evolution of the system produces complex behaviours and solutions [112, 117]. Intuitively, large-scale, long-term interactions would play a similar role as size (the so-called "scaling laws for LLMs" [52]) in shaping complex behaviour and dynamics that would not emerge otherwise.

Currently, despite growing hype around the potential of LLMs in MAS contexts, which often exceed their demonstrated abilities [98], we foresee the risk of a birth and death of interest in emergent behaviours in LLMs. Emergence alone, primarily when arising from loosely constrained prompts and undefined interaction dynamics, further compromised by hallucinations and memory issues in LLMs, does not justify claims of MAS-level coordination or long-horizon planning. These factors make distinguishing between genuine coordination and coincidental or spurious outputs increasingly difficult.

We therefore argue that the evaluation of emergent behaviours in LLMs should rely on quantifiable metrics rooted in established MAS [8, 33, 56, 84].

We elaborate on these concerns and devise a research path in Section 5.

---

In summary, our position in this paper is that:

> **Paper Position**
>
> Current **MAS-LLMs often fail to embody fundamental multi-agent system characteristics**, such as autonomy, social interaction, and structured environments, **by overemphasising the role of LLMs and overlooking solutions that already exist in MAS literature.**

## 2 Social Intelligent Agents: LLM Agents Lack Native *Social Behaviour*

In MAS, an agent is commonly defined as intelligent if it is reactive, proactive, and shows social behaviours [94, 123]. While LLMs are reactive and proactive, this section discusses why they often fail to be social agents and where this discrepancy originates.

For any agent, the prerequisite of reactiveness is perception, i.e., interpreting an input to take an associated action. LLMs are reactive agents, i.e., respond to stimuli from a changing environment, with the input being directly injected into the prompt [99, 105, 132] or retrieved via external information [51, 109].

The proactiveness of an LLM is its capacity to initiate tasks with limited or without human intervention [72]. Independence eventually arises in LLMs only when the model can self-generate the prompt or self-inject information [82]. Recent works have focused on proactive LLMs that ask clarification to the user on the instructions provided, retrieve additional information from external resources or are fine-tuned to anticipate the next actions [136, 139].

The third intelligent characteristic of an agent is the capacity to behave socially, thus to compete or cooperate [21, 124]. Competition and cooperation arise from the capacity to be both reactive and proactive, i.e., react to other agents' actions and initiate new ones. Crucially, agents must be socially reactive and proactive, i.e., able to grasp and reason about other agents' goals, negotiate with them, and even enlist their cooperation when needed.

The literature on LLMs includes a substantial body of works aimed at developing competitive and cooperative systems of LLMs [71]. In most of them, the agents' roles are scripted through prompting [29, 79] or fine-tuned [73, 108, 130], but not natively trained to cooperate or compete with one another.

A recent survey identified that 37% of the failure cases of MAS LLMs are errors caused by inter-agents misalignment or agents' coordination issues [14] (Figure 2). In the same spirit, [62] shows how multi-agent reinforcement learning agents outperform LLMs at planning tasks, highlighting the role of pre-training. Concurrently, a growing body of research in Machine Theory of Mind (ToM) [89] evidences how LLMs struggle to express their beliefs, desires, and intentions [27, 91], and those of other agents [106, 115, 116].

Furthermore, when multiple LLMs interact, their specialisation, achieved through different prompts or fine-tuning, often aims to maximise a single objective such as accuracy, at the expense of other desirable behavioural properties. As a consequence, several MAS LLMs reduce to aggregation mechanisms like majority vote [30, 53, 65, 107].

The lack of native social behaviour, the tendency to converge to ensembles of LLMs rather than developing concurrent strategies, and the well-known issues of LLMs with ToM, contribute to not making LLMs natively interactive agents; in most agentic tools,[3] the *environment* makes them cooperative (e.g., via an orchestrator, workflows, or by prompting them with each other's output).

To summarise, this section points out that most frameworks rely on orchestrators and workflows to direct the LLMs' behaviour, with interactions that occur as a by-product of initial instructions. From a MAS perspective, LLMs reactivity and proactiveness are not socially directed. Instead, in the following paragraph, we argue that social agents should be natively trained to interact, collaborate, and cooperate with other entities such as LLMs, humans, other algorithms and tools, etc.

**Research directions.** We argue for LLMs whose pre-training phase encompasses cooperation and competition in different scenarios. As LLMs are trained on textual corpora to approximate the distribution of human language, we should consider teaching agents the basics of multi-agent cooperation and competition.

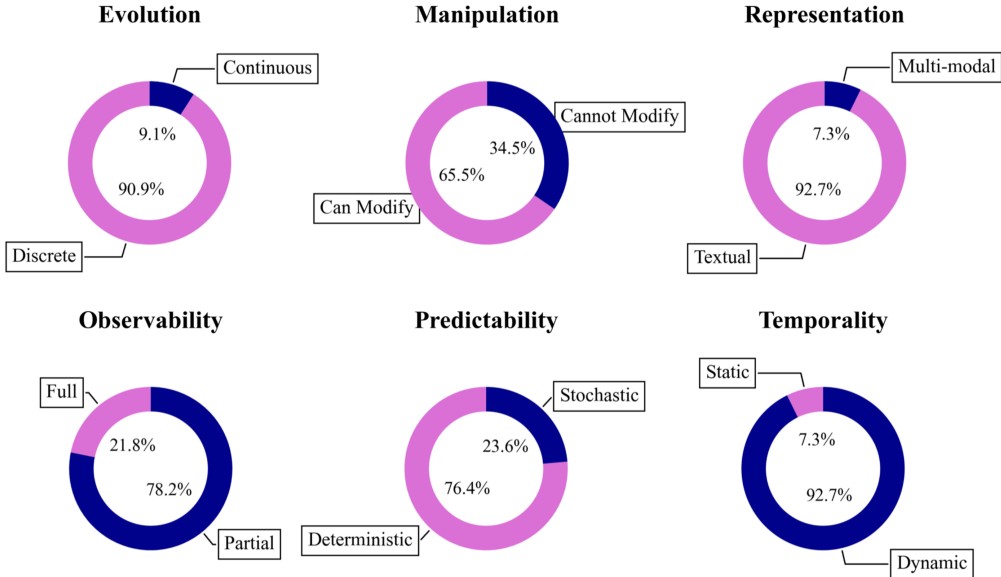

Figure 1: We categorise the environments of approximately 100 MAS LLMs Benchmarks & Evaluations papers published between 2023 and 2025. For those papers that describe or allow inference of their environment characteristics (Section 3), we present the data using wheel plots. See the papers list in Appendix A.

Recent advances in the field allow training LLMs directly on textual feedback they receive from other models [135]; that, alongside cooperative or competitive reinforcement learning, where agents are trained not only to complete tasks but also to adapt to and influence each other's behaviour, can serve as a path towards MAS in which LLMs show reactive and proactive behaviours beyond answering to prompts. An agent trained in this way would learn to respond appropriately while also interpreting, anticipating, and reacting to the actions of other agents, based on feedback from their interactions.

In conclusion, while fine-tuning is proven to be promising to specialise and assign roles to LLMs [68, 73, 108], we argue it is alone insufficient to provide them with the capacity to cooperate and compete.

# 3 Environment Design: MAS LLMs Environments are LLM-centric

As anticipated in the Introduction, MAS traditionally model the environment with few assumptions regarding the architectures of the agents that will populate it. On the other hand, MAS LLMs centre around LLM-powered agents, overlooking the interoperability with agents that do not conform to them. Furthermore, the intrinsic limitations of LLMs, such as their proneness to hallucinations [69, 143], lack of determinism, and long-term memory persistency [83] may hinder the development of the field, in particular in scenarios where safety and time (as measured in seconds [123]) are paramount.

In MAS, an environment is the external context that agents perceive and react to, and can be characterised across five dimensions [94, 122]. *Observability* determines the amount of information an agent has about the environment and other agents. *Predictability* states an environment's determinism level (from fully deterministic to stochastic). *Temporality* concerns the dependency between successive environment states. *Evolution* determines how the environment changes with time (i.e., from discrete or continuous). *Manipulation* represents the degrees of freedom agents have in interacting and modifying the environment.

Of around 110 MAS LLMs articles published between 2023 and 2025 and whose results we summarise in Figure 1, most MAS LLMs operate in partially observable, deterministic, temporally

dynamic, discrete environments [66, 77] (Further details on the papers in Appendix A). Furthermore, the vast majority of LLMs articles (see Figure 1, bottom-right) employ textual representations.[5]

In a fully observable setting, agents receive the same, shared representations of the environment; conversely, partially observable environments provide partial, unique representations to each agent. In MAS LLMs observability, whether partial or full, comes with some issues due to the nature of LLMs: in fact, these models notoriously fail at inferring other agents' beliefs, desires, and intentions [97, 100, 115]. As discussed in Section 2, LLMs tend to integrate all the information they have access to without distinguishing what they *know* as opposed to what they *know what other agents know*, the so called problem of $k^{th}$ order beliefs [87]. In this sense, a centralised MAS LLMs where an LLM orchestrates and monitors the others may incur issues regarding attribute *who did what* or *what an agent wants to achieve*. Even systems with two LLMs suffer from these issues [61], as we later discuss in the last paragraph of this section.

While most MAS LLMs papers assume the environment is deterministic and can be modified (i.e., it changes according to the actions the LLMs make), the intrinsic non-determinism of LLMs flaws this setting. In terms of safety, one can design specific procedures, such as safety mechanisms and guardrail measures, that deterministically trigger when particular events happen; on the other hand, non-deterministic LLMs provide no guarantees they will behave accordingly [20]. Even setting the temperature of an LLM to zero or sampling deterministically their strategies is not a solution, as many LLMs are known to behave always non-deterministically [58, 83].

In two popular frameworks, Camel [61] and MetaAgent [67], the authors illustrate how their models hallucinate and underperform when MAS LLMs are specialised to perform a subtask. In Camel, two LLMs autonomously generate prompts to solve complex tasks, reducing reliance on user input. The authors notice that LLMs can inadvertently swap their roles, generate repeated or non-useful instructions or get stuck in infinite message exchanges. MetaAgent focuses on collaborative agents that accomplish coordination tasks. Through perception, memory, reasoning, and execution modules, LLMs interact with the environment, store valuable information, and learn rewarding skills. The authors show that LLMs deviate from their predefined identities, hallucinating their competence and thus compromising collaboration.

We argue that these problems are a byproduct of the overreliance on natural language and free-text as the reference medium for coordinating agents. Natural language is inherently ambiguous and prone to misinterpretation [10, 64], suffers from information loss [62], and requires high costs for storing and retrieving information. In the following paragraph, we propose some research directions to mitigate these issues.

**Research directions.** We propose the following research directions to address the issues with textual environments. First, we argue it is crucial to explore multi-modal environments where the stimuli are turned into actionable steps without intermediate translation into natural language [49, 70, 137]. The intuition is that the fewer mediations a stimulus goes through, the less likely the signal will contain distortions or introduce errors.

Furthermore, using structured formats (taking inspiration from performative agentic languages such as KQML [36]) can remove the ambiguities inherent in natural language communication. Last but not least, integrating LLMs and formal planners or neuro-symbolic methods can provide guarantees that precise actions will be carried out correctly: LLMs excel at extracting a task's specifics from noisy and incomplete specifications, while formal methods can provide plans that are guaranteed to achieve the goal.

## 4   Coordination and Communication Issues in MAS LLMs

This section addresses the lack of asynchronicity and standardised communication methods.

**Coordination and the lack of asynchronous MAS LLMs.** As described in Section 1, asynchronicity is a key component of multi-agent concurrent systems: in its basic form, concurrent systems encompass multiple, diverse tasks executed simultaneously, without assuming when they start or

---

[5]As communication systems are not intrinsically part of the environment, we further elaborate on the non-sustainability of text as the preferred medium of communication in large networks of MAS LLMs in Section 4.

end. Examples of notorious problems that can only be modelled with asynchronicity are those that necessitate concurrent algorithms to be solved [24, 26]: these include classic problems such as the "dining philosophers", handshaking protocols, etc. Asynchronicity also arises in many practical scenarios, such as email and chat exchanges and database management access. Many real-world scenarios cannot be modelled without asynchronicity and require simplification (e.g., by assuming agents act sequentially through an orchestrator).

Notably, while most closed-source LLM providers offer asynchronous APIs, agents employing LLMs tend to be predominantly used in synchronous or parallel fashion [65].

In this sense, we surveyed the MAS LLMs literature published between 2022 and 2025, to understand how many works directly employ asynchronicity or enable the deployment of asynchronous agents. We identified few works (22) that explicitly model or discuss asynchronous agent interactions.[6] Furthermore, in those few cases, asynchronous interactions are implemented through conversations and by employing frameworks and languages that are not natively asynchronous, which adds unnecessary complexity and reduces interoperability. More information is provided in Appendix B.

For example, we consider the case of AutoGen, an influential MAS LLMs framework [127] that enables building LLM applications through multiple interacting agents. Although AutoGen supports asynchronous calls,[7] developers must define asynchronous calls for each action and event; asynchronous programming with synchronous languages is well-known to be prone to bugs that impede the system from being fully asynchronous. As we expand later, we argue for a reverse approach. Every agent and environment should be natively asynchronous to be considered as a MAS,[8] with sequential calls being the exception, rather than the norm.

**Communications methods in MAS LLMs.** Traditionally, there are three levels at which (Multi-Agent) communication is analysed [6]. An utterance that conveys some meaning may have to the hearer no intended effect (an *illocutionary act*, e.g., "the sky is blue"), some meaning intended to warn (a *perlocutionary act*, "a train is passing"), or some meaning that acts as a request (a *performative*, e.g., "please open the window"). These distinctions, alongside the so-called *rules of conversation* (implicature, the Gricean maxims [41], etc.), constitute the foundation of agents' communication in MAS [11, 48, 96].

In MAS, illocutionary conversations are usually handled using descriptive and structured languages, such as JSON and RDF. Their purpose is to exchange information, not to perform or request actions. On the other hand, KQML [36], FIPA's ACL,[9] and rational programming and Agent-Oriented Programming [103] are examples of consolidated *performative* standards that describe agents' beliefs, commitments, and actions. For instance, for an agent that exchanges a file with another and asks to summarise it, the MAS literature proposes using a structured language for the exchange and a performative query for the summarisation.

When it comes to LLMs, humans interact with them in free-text form; by extension, most MAS LLMs systems adopt natural language as the primary communication medium. While natural language captures the nuances of the principles mentioned above, it comes at the cost of complexity and ambiguity: an utterance such as "a car is coming" may range from being *illocutionary* (conveying information) to *performative* (conveying an implicit warning or an order). A few recent works in MAS LLMs propose to handle handshakes and errors with natural language communications; any other communication that routines and protocols can implement should otherwise favour structured languages [76].

To conclude, overlooking the importance of communication by assuming natural language as the standard has the concrete risk of developing MAS LLMs that are expensive (the cost of generating responses would dwarf any other in the system) and where language ambiguities cause failures that are hard to inspect, fix, and prevent.

**Research directions.** In terms of asynchronicity, we devise two complementary research directions that are worth investigating: on the one hand, frameworks that model MAS as asynchronous sys-

---

[6]As a reference, this survey counts more than 1400 MAS LLMs articles: `https://github.com/AGI-Edgerunners/LLM-Agents-Papers`

[7]`https://github.com/microsoft/autogen?tab=readme-ov-file#web-browsing-agent-team`

[8]This core characteristic is discussed in detail in [123], Chapter 1.3.

[9]`http://www.fipa.org/specs/fipa00061/XC00061D.html`

tems [22] should be adapted to MAS LLMs; the capacity of modelling asynchronous systems can provide insights and guarantees into their long-term evolution. One example is that described in [1], which models MAS with Petri nets, a model of computation that is inherently asynchronous, enabling analyses on reachability, boundedness and invariance of the system.

On the other hand, as mentioned above, providers often expose asynchronous LLM APIs, for which we should develop suitable environments. Critical points in this sense are how deadlocks and starvation are handled to ensure a consistent evolution of the environment [39, 95]. To summarise our position, we argue for MAS LLM frameworks that are natively asynchronous and where sequential actions are the exception. As regards communication systems, we argue for standard, open-source frameworks to reason and build the key components of any LLMs communication (e.g., aspects such as security, identity preservation, trust, message exchange, etc.).

In line with some recent initiatives,[10] we believe the MAS LLMs community should work towards standard agent protocols guided, where needed, by MAS principles. In the spirit of Agent Oriented Programming [121], researchers would have abstract templates that already implement security routines (e.g., a communication between web agents would only happen over a secure channel such as HTTPS).

In summary, we argue for communication systems between LLMs built on top of what the computer science and the MAS community consider good practice and standards regarding security, identity preservation, trust, handshakes, etc.

## 5 MAS LLMs do not Quantify the Emergence of Complex Behaviours

In many disciplines, such as system theory, economics, MAS, etc., an emergent behaviour is observed when a complex entity has properties or behaviours that its parts do not have on their own, and emerge only when they interact in a wider whole [4, 37]. In the context of LLMs, emergence describes the increasing capabilities of a model at varying model size [12, 52]. With MAS LLMs, behaviours that cannot be predicted via a static analysis of agents and their environment are also considered emergent [62, 85].

Emergent behaviours in MAS and MAS LLMs are often associated with open-ended environments, i.e., those MAS settings where LLMs interact and evolve freely. In an influential work [85], Park et al. use LLMs to simulate a sandboxed society, with a focus on how single instructions influence the population and information spreads across the environment. While the authors show that simply nudging one agent causes other agents to engage in complex behaviours, the concept of emergence itself is never addressed formally.

Another recent work studies how LLMs build agent societies within a Minecraft environment [3]. While the work claims that agents can achieve significant milestones towards AI civilisations,[11] results are primarily observational, i.e., the system is let evolve for a long time and then researchers make their observations on interesting behaviours the LLMs adopt. Other works approach emergence from a similar perspective [85, 128], and a body of research studies emergence before the advent of ChatGPT [40, 59].

We thus surveyed papers published in the MAS LLMs and machine learning community between 2023 and 2025 that mention emergent behaviours to understand what methods and metrics employ to identify and quantify them. Out of more than 60 papers analysed, only a few define clear metrics to measure emergent behaviours, while the majority qualitatively evaluate such behaviours and report the most notable. More details about the analysis and the list of papers are reported in Appendix C.

In light of these observations, we question whether the behaviours described in these works are natural outcomes of the actions of powerful general-purpose LLMs, as discussed in Section 2, or truly represent emergent behaviours.

In conclusion, in this body of research, we observe (i) no systematic definition of what constitutes an emergent behaviour, without reference to the system being analysed, therefore (ii) a lack of proper benchmarks to quantify them. In contrast, traditional MAS research generally insists on rigorously defined environments, quantifiable objectives, and formal verification methods. Emergence alone,

---

[10]https://github.com/google/A2A, https://nanda.media.mit.edu/, https://las-wg.org/

[11]A video showcasing their emergent behaviours, https://www.youtube.com/watch?v=9piFiQJ-mnU.

especially when derived from loosely constrained prompts and undefined interaction dynamics, does not suffice to claim long-horizon MAS-level coordination or planning capabilities [114].

**Research directions.** We argue for a proper definition of emergence and emergent behaviours in open-ended MAS LLMs, where the concept is well established and scientifically falsifiable [4, 37].

For example, in economics, emergence is sometimes characterised in its core aspects, and encompasses behaviours that produce outcomes the theory can explain [57]: in other words, emergent phenomena are *economics* phenomena. In this sense, since MAS LLMs share with economics the interest in agents' behaviour, a characterisation of emergence can encompass phenomena that relate to the agents' objective functions (e.g., in a system where agents have to maximise productivity, one finds a way to hack the reward function).

Conversely, when emergent phenomena are beyond the scope of the system (e.g., in a system where agents have to maximise productivity, another objective function spontaneously emerges), that would represent a different facet of emergence.

While this definition has the advantage of being measurable, it fits the notions of weak and strong emergence in computer science [15] (a weak emergent phenomenon can be derived from the underlying system; a strong phenomenon requires new laws or assumptions), thus making the adaptation to MAS LLMs more straightforward.

# 6   Alternative Views

**Section 2 - MAS LLMs do not need social pre-training to be social agents & Section 4 - Central orchestration is enough to build complex MAS LLMs.** The main argument against our position in Section 2 and the first paragraph of Section 4 are that (i) agentic tools do not need to pre-train LLMs to enhance their *social behaviour* and capabilities to interact with other agents and that (ii) simple orchestrators and agentic workflows suffice to coordinate complex MAS LLMs interactions.

Implementations of MAS LLMs tools[3] propose workflows orchestrated through predefined code paths, with LLM-powered agents maintaining control over how they accomplish tasks. Human-designed workflows orchestrate coordination, as well as the general scope of intra-agent iterations. Google also published its agentic framework,[3] which puts emphasis is given to how agents perform their tasks leveraging existing techniques such as ReAct, Chain of Thought, Tree of Thought, etc. [120, 133, 134], and no mention to the social aspects, and the relative training, of agent interactions.

Similarly, Microsoft AutoGen [127] (which we discussed extensively in Section 4) and OpenAI Agents [101],[3] do not mention the possibility to pre-train agents to develop *social behaviour*, as they see agents as components that solve a task managed by the orchestrator. Finally, the debate around whether LLMs possess *social behaviour* and a Theory of Mind has prominent supporters [55], whose arguments are strong but mostly empirical [106].

**Section 3 - MAS LLMs environments are not LLM-centric.** The main concurrent arguments to our point in Section 3 are that (i) most practical scenarios do not require open environments [123] and that (ii) progress in the field will overcome their limitations regarding lack of stable memory, hallucinations, and the costs of storing and retrieving information in (textual) natural language.

Regarding point (i), influential work from industry underscores the importance of avoiding overly complex frameworks for MAS LLMs, advocating for simple, modular design principles [14]. These principles are adopted by Anthropic, Google, Microsoft, and OpenAI frameworks, enabling LLM capabilities testing in open-ended environments. However, while the tools to construct such challenging scenarios exist, the community has shown limited practical interest in doing so. Instead, current research efforts focus more on exploring what can be built with MAS LLMs than on stress-testing them in challenging settings.

As regards point (ii), many recent works address the problem of equipping LLM agents with long-term memory [118], retrievable with reduced computational costs [50], as well as methods to reduce hallucinations [34, 35] while maintaining text as the primary input and communication medium.

**Section 4 - Basic communication systems are enough to build complex MAS LLMs.** The main concurrent argument to our point in Section 4 is that natural language is sufficient for large-scale communication in MAS LLMs.

Some works in the literature have already identified scaling issues in MAS LLMs, and have proposed potential countermeasures [119, 138]. In [138], a reduction in token usage is achieved by 28-73%, but requires an expensive RL-based optimisation of the communication graph that does not transfer across tasks. Similarly, in [119], an average 22% reduction in prompt tokens and an average 18% reduction in completion tokens is achieved, but requires optimising a set of parameters that is not transferable across tasks.

**Section 5 - The point of open-ended MAS LLMs is not benchmarking.** The main concurrent argument to our point in Section 5 is that emergent behaviours are all those systems that exhibit, in the long run, complex and often surprising capabilities that were not programmed or predicted during development.

This phenomenon of emergence is not unique to artificial intelligence; it is well-recognised across other disciplines. In biology, for example, the flocking of birds or the organisation of ant colonies are classic emergent phenomena, i.e., complex patterns arising from simple rules followed by individuals [13].

Similarly, in economics, market trends and crashes often emerge from the interactions of countless agents acting on local information [5]. In these fields, emergent behaviours are typically treated as observational phenomena recognised through empirical study mirroring how they are now being explored in machine learning [78].

# 7 Conclusion

Our work argues that the current literature on MAS LLMs overlooks key issues already explored in the MAS literature. First, LLM agents currently lack native social behaviour; We think that LLMS can be pre-trained to learn cooperation and competition through multi-agent scenarios and interactive feedback, enabling them to develop socially adaptive behaviours beyond prompt-based responses. Second, MAS LLMs environments are overly centred on LLMs themselves; We argue for prioritising the development of LLMs capable of accessing the current state of their environment, through structured memory systems, without relying on their memory. In parallel, we support the design of external reward mechanisms that can reliably assess LLMs performance in multi-agent settings. Third, MAS LLMs lack asynchronicity and over-rely on natural language as the primary communication protocol; We propose developing MAS LLM frameworks that are natively asynchronous and grounded in standardised, open-source communication protocols, ensuring security, identity, and trust, drawing from established practices in multi-agent distributed systems and communication. Finally, while emergent behaviour is cited as a desirable property in MAS LLMs, it lacks a rigorous definition in this context. We suggest establishing a clear, falsifiable definition of emergent behaviours in open-ended MAS LLMs, aligning with how emergence is rigorously treated in MAS.

# Acknowledgments

ELM was supported by the Alan Turing Institute. GLM was supported by UK Research and Innovation [grant number EP/S023356/1], in the UKRI Centre for Doctoral Training in Safe and Trusted Artificial Intelligence (www.safeandtrustedai.org). SM was supported by the EPSRC Centre for Doctoral Training in Autonomous Intelligent Machines and Systems n. EP/Y035070/1, in addition to Microsoft Ltd. MW was supported by an AI 2050 Senior Fellowship from the Schmidt Sciences Foundation. ELM, SM, and PT are affiliated with the Institute for Decentralized AI, which they thank for its support.

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

## A  Categories of Environments

The 112 papers used to produce the results in Figure 1 are listed in this repository, `https://github.com/AGI-Edgerunners/LLM-Agents-Papers?tab=readme-ov-file#Infrastructure`, last access May 9th 2025. They account for the most influential and popular MAS LLMs "Benchmark & Evaluation" papers published between 2023 and April 2025. We focus on the category "Benchmark & Evaluation" as it encompasses new benchmarks built to test the capabilities of LLMs in Multi-Agent settings, as well as extensive evaluations.

## B  Papers that Leverage or Study Asynchronicity in MAS LLMs

Below are the papers surveyed to understand how many works, between 2023 and 2025, explicitly leverage asynchronicity to model complex scenarios. The list includes academic papers (e.g., arXiv, conferences like NeurIPS, ICLR, ICML), major industry reports, and whitepapers.

**Frameworks and Architectures for Asynchronous Multi-Agent LLMs.**   [18, 44, 61, 127, 142]

**Applications of Asynchronous Multi-LLM Systems.**   [2, 7, 9, 16, 25, 28, 29, 45, 46, 47, 63, 74, 75, 80, 85, 129, 141]

## C  Papers on Emergent Behaviours and Their Measurement in MAS LLMs

The papers used for the survey in Section 5 come from sources like Google Scholar, Scopus, and this survey `https://github.com/tsinghua-fib-lab/LLM-Agent-Based-Modeling-and-Simulation`.

