# OpenReview forum: "Large Language Models Miss the Multi-agent Mark"
_NeurIPS.cc/2025/Position_Paper_Track — NeurIPS 2025 Position Paper Track_

### Official Review · Reviewer_uLzx · 2025-08-04

**Significance:** 3
**Presentation:** 3
**Rating:** 5
**Confidence:** 3

**Summary:**

This paper argues that current research on Multi-Agent Systems of LLMs (MAS LLMs) is missing foundational concepts from traditional MAS literature. In particular, the authors identify four key shortcomings of current MAS LLM research and advocate for improving them by bringing LLMs into traditional MAS research:
1. LLM agents lack social intelligence, like interactive capabilities.
2. MAS LLM environments are LLM-centric and communicate via natural language.
3. Coordination and communication protocols ignore issues studied in MAS research, e.g., asynchronous design.
4. Emergent behaviors are claimed without rigorous definitions or benchmarks.
The authors advocate for integrating MAS principles into LLM-agent system design and highlight future research directions, such as pretraining LLMs for social intelligence and using asynchronous frameworks.

**Strengths:**

- The paper studies a well-motivated topic that is relevant to the NeurIPS community. MAS is the next step for AI research, while traditional MAS research is overlooked in many cases.

- The authors seem to have good awareness of both traditional MAS and LLM agent literature.

- The paper raises timely concerns about important issues, including designs like coordination protocols, evaluation standards, etc.

- The paper provides some future research directions for each limitation it points out.

**Weaknesses:**

- The support of some arguments is not super strong. The position may be stronger if it is substantiated through more concrete comparative analysis to show how traditional MAS can help more specifically.

- Some recommended future directions are vague. Ideas like pretraining LLMs to be social or adopting MAS-style communication are suggested, but how feasible these directions are is not entirely clear or concrete.

- Some of the alternative views are addressed, but some are not.

**Questions:**

- The paper proposes pre-training LLMs for social intelligence by exposing them to multi-agent interactions and behaviors. However, if the dataset already captures such interactions, whether from humans or other models or any sources that dataset is generated, isn’t that sufficient? What additional benefit does explicit multi-agent pretraining offer?

- How can we validate whether an LLM has truly learned “social behavior” or developed true Theory of Mind in a MAS setting?

- The authors argue that natural language is inefficient for agent communication, yet humans coordinate effectively using it. Are the authors suggesting that the development of MAS LLMs should not ultimately converge to human-like communication?

**Alternative Position:**

Yes, and alternative positions are well-considered and named but not addressed

**Author Identification:**

No.

**Context:**

2

**Discussion:**

3

**Ethics:**

["NO or VERY MINOR ethics concerns only"]

**Position:**

Yes, the paper argues for or against a position related to machine learning.

**Support:**

2

**Thoroughness:**

3

---

### Official Review · Reviewer_oQRJ · 2025-08-07

**Significance:** 4
**Presentation:** 4
**Rating:** 9
**Confidence:** 4

**Summary:**

The position paper presents an important argument relevant to the development of LLMs in multi-agent systems (MAS). It highlights that many current MAS-LLM approaches lack essential multi-agent characteristics such as autonomy, social interaction, and structured environments, and instead rely on overly simplified, LLM-centric architectures. The paper articulates its position through four core critiques: (1) the absence of native social behaviors, (2) the dominance of LLM-centric environments that do not reflect realistic MAS settings, and (3) the insufficient treatment of coordination and communication challenges, (4) in quantifying the emergence of complex behaviors. To support these claims, the authors draw on compelling evidence from existing literature and methods.

**Strengths:**

The paper effectively challenges the dominant LLM-centric design paradigm and encourages a shift toward more principled, socially grounded MAS research.

The argument is clearly communicated, and grounded in evidence from the literature, making it a valuable contribution to ongoing conversations at the intersection of ML, AI, and multi-agent research.

Furthermore, the paper is well-organised, with strategically placed side notes that effectively highlight key points and enhance readability.

**Weaknesses:**

Minor problem:
1.  The font size in Figure 1 is too small to read.

**Questions:**

The paper references a substantial body of literature to support its arguments. Incorporating summary figures or tables to synthesize and categorize these works could enhance the clarity and readability of the paper. Is it possible?

**Alternative Position:**

Yes, and alternative positions are well-considered and addressed by the argument

**Author Identification:**

No.

**Context:**

4

**Discussion:**

4

**Ethics:**

["NO or VERY MINOR ethics concerns only"]

**Position:**

Yes, the paper argues for or against a position related to machine learning.

**Support:**

4

**Thoroughness:**

3

---

### Official Review · Reviewer_pTLG · 2025-08-28

**Significance:** 3
**Presentation:** 3
**Rating:** 6
**Confidence:** 3

**Summary:**

The paper argues that the recent development of LLM agents ignore decades of literature on agentic systems.

**Strengths:**

The paper argues that MAS-LLMs do not have the characteristics of traditional multi-agent systems. For each of the four characteristics, the authors give evidence from either first principles or from the literature that the MAS-LLMs fail to have a particular characteristic. The paper also gives suggestions for improving the MAS-LLMs by following in the prior MAS literature.

**Weaknesses:**

Not a weakness per se, but a nit. I would have liked to see the paper be organized to be less repetitive. As it stands, the position doesn't appear until the bottom of page 3 because the key portions of the arguments from sections 2-3 are outlined in the introduction at 4-5 paragraphs for each argument. This is excessive; I would have preferred to get to the position faster. The additional space could have been used to clarify the methodology of the literature review etc etc.

Additional nit: the text of figure 1 is illegible without breaking out my reading glasses. Please be kind to the aged!

**Questions:**

1. I believe that the LLM literature does not appropriately follow the prior literature. Why is that important? The paper currently reads as if not following the literature was simply a thing to be avoided, without arguing *why* not following the literature is bad for this specific case. (Other than a brief mention of "slowing down" or "losing traction".

**Alternative Position:**

Yes, and alternative positions are well-considered and addressed by the argument

**Author Identification:**

No.

**Context:**

3

**Discussion:**

3

**Ethics:**

["NO or VERY MINOR ethics concerns only"]

**Position:**

Yes, the paper argues for or against a position related to machine learning.

**Support:**

3

**Thoroughness:**

3

---

### Note · Authors · 2025-08-25

**1-11 Submit Again:**

Probably yes

**1-1 Submission Process:**

4

**1-2 Next Year:**

A clearer breakdown of deadlines and the release of the reviews would be appreciated.

**1-3 Future Development:**

We like the current idea behind the review process and we support it; however, having the opportunity to have a rebuttal with the authors of the reviews, as in the main conference, would be ideal.

**1-4 Interest:**

["Panel discussions with other position paper authors", "Structured debates on controversial topics", "Mentorship programs for early-career researchers"]

**1-5 Thoughtful:**

7

**1-6 Supportive:**

8

**1-7 Technical Aspects Versus Position:**

9

**1-8 Gate Keeping:**

10

**1-9 Camera Ready Changes:**

We would like first to address the questions reviewer uLzx raised. In particular, we expand on the social pre-training idea. Our position is that in the same way next-word prediction models language, learning social skills while learning language may ground concepts and have the same effect as multi-modal training in bridging the gap between text and images [1]. In other words, we propose to treat social intelligence as a modality. We believe this could lead to a breakthrough in the field of Multi-agent LLMs and, while a detailed discussion of our idea would require a dedicated, technical paper (going thus beyond the scope of the position track), we are happy to further expand on our ideas and vision in the final version of the article.

We will also provide more details about the issues of natural language as a communication system for LLMs. While we already cite the literature on the limits of natural language for LLM communication, the comparison with humans and their use of natural language as a standard communication medium can be further detailed. Our position, which aligns with the current MAS-LLMs literature, is that the comparison between humans' and machines' communication is a red herring [2]: humans' characters and alphabets are signs, while LLMs receive high-dimensional vectors and never manage to see the correspondence between a token and its representation. This makes the two communication media very different, and comparisons should be drawn with caution. We are happy to discuss our position further.

We finally notice that reviewer oQRJ gave us a very high score (9) with high confidence, and did not point out any concern about the position of our work. They suggested adding a few more figures and making the one present more readable. We commit to making these changes in the final version of the paper.

[1] How to Bridge the Gap between Modalities: Survey on Multimodal Large Language Model, TKDE
[2] Language Models are Implicitly Continuous, ICLR'25

**3-1 Review Response1:**

uLzx

**3-2 Reaction To Review1:**

We thank the reviewer for highlighting the strengths of our paper. The authors say that our paper would positively contribute to the debate around a central topic in LLM research. Still, they are concerned about the future directions and the mitigations that we propose to existing problems in MAS-LLM systems.

We thank them for raising these points: we already discussed how we intend to address these points in the Section “Camera Ready Changes” (above).
We kindly point out to the reviewers and the AC that part of the survey.

Here, we would like to discuss two points related to the review itself.

1. We intended to first clearly state that we believe there is a misalignment between LLMs, agentic, and MAS research; we are glad to see that our four points and the position were well-received and appreciated by all the reviewers.

On the other hand, we are aware that some researchers may disagree with our proposed research directions. While we believe that social pre-training can lead to a breakthrough in the field of MAS-LLMs (see “Camera Ready Changes” for more details about our vision),  our primary intent was to delineate some research directions that could inspire other researchers, as well as fuel a dialogue in the community. Given their nature, i.e., they address a big open problem in the field, the research directions that we propose are intrinsically contentious as they channel a large amount of research in a direction, automatically subtracting it from others.

2. We are happy that the reviewer found the article interesting for the broad community at NeurIPS; In this sense, we would like to ask the reviewer a question: we reckon that their review is positive (see Strengths vs. Weaknesses), and we wonder if the numerical score (5, borderline accepts) balances the strengths and weaknesses as expressed in the textual evaluation. If that is not the case, we’d be grateful if they considered aligning their score with the positivity of their review!

**3-3 Review Response2:**

oQRJ

**3-4 Reaction To Review2:**

We thank the reviewer for appreciating our paper. We are delighted they found our position, presentation, and future directions compelling and of interest to the broad NeurIPS community. Indeed, we believe this article can spark a lot of interesting discussions at the conference! We are also happy to fix the issues they spotted with the presentation (make Figure 1 bigger, add some more figures to the paper).

**3-5 Review Response3:**

pTLG

**3-6 Reaction To Review3:**

We thank the reviewer for the review, and we are glad they confirmed the positive impressions of the other two reviewers.

As regards the first nit/weakness, indeed, the position comes quite late in the introduction. This choice was discussed between the authors and has a rationale behind it; since the position expressed in the paper is strong and potentially divisive, we gave as much background as possible before stating it. We are happy to make the introduction lighter and present the position earlier.

As regards the second nit/weakness, we will improve the size and quality of the plot in the final version of the article.

Reviewer pTLG also asks the following question:

- *I believe that the LLM literature does not appropriately follow the prior literature. Why is that important?*

This is a fair point: we will argue more strongly that ignoring or not following design patterns that are well established in the MAS literature comes with the risk of developing brittle systems, beyond slowing down the research. In this sense, one analogy we can draw, albeit with some relevant caveats, is with the anti-patterns issue in software engineering [1], i.e., solving problems with apparently consolidated yet flawed patterns.

There is already strong evidence in the literature that many successful MAS-LLM frameworks overlook critical aspects of MAS. Inefficient communication systems, poor security, and a lack of native asynchronicity are hazardous because successful MAS-LLM tools can reinforce anti-patterns that become increasingly difficult to eradicate.
The MAS literature already offers consolidated patterns and solutions to the issues mentioned above. One example in the paper is that of the MAS performative standards that describe agents’ beliefs, commitments, and actions [2], which can be used to make the communication between agents unambiguous and safer.

[1] Budgen. Software design, 2003
[2] Multiagent Systems: Algorithmic, Game-Theoretic, and Logical Foundations.

---

### Meta-Review · Area_Chair_1Sth · 2025-09-17

**Rating:** 6
**Confidence:** 3

**Strengths:**

The paper addresses a well-motivated and timely topic that is highly relevant to the NeurIPS community, especially given the growing interest in multi-agent systems (MAS) and LLMs.

It clearly argues that MAS-LLMs lack the defining characteristics of traditional MAS, supporting this claim with evidence from first principles and prior literature. The position challenges the dominant LLM-centric paradigm and encourages a shift toward more principled, socially grounded MAS research.

The paper demonstrates the authors’ strong awareness of both traditional MAS and LLM-agent literature, allowing for a balanced perspective.

Suggestions for improving MAS-LLMs are well-connected to prior MAS research, with some concrete directions outlined for future work.

**Weaknesses:**

The organization is somewhat repetitive, large portions of the arguments from sections 2–3 are restated in the introduction, delaying the core position until later in the paper.

Some arguments would benefit from stronger empirical support or comparative analysis to concretely demonstrate how traditional MAS approaches could improve MAS-LLMs.

Several proposed future directions are described at a high level but lack specificity or feasibility details. Some alternative perspectives are acknowledged, but others are insufficiently addressed, limiting the breadth of the discussion.

Minor issues: Figure 1 font size is too small to be easily legible.

**Questions:**

The paper suggests pretraining LLMs for social intelligence via multi-agent interactions. If training data already contains such interactions (from humans, models, or other sources), what additional benefits does explicit multi-agent pretraining provide?

How can we rigorously validate whether an LLM has genuinely acquired “social behavior” or Theory of Mind in a MAS context?

The authors argue that natural language is inefficient for agent communication, yet humans coordinate effectively with it. Do the authors believe MAS-LLMs should ultimately avoid converging toward human-like communication, or instead refine it?

**Thoroughness:**

3

---

### Decision · Program_Chairs · 2025-09-26

Accept